# *Caenorhabditis elegans* as a Model for the Effects of Phytochemicals on Mitochondria and Aging

**DOI:** 10.3390/biom12111550

**Published:** 2022-10-24

**Authors:** Fabian Schmitt, Gunter P. Eckert

**Affiliations:** Laboratory for Nutrition in Prevention and Therapy, Biomedical Research Center Seltersberg (BFS), Institute of Nutritional Science, Justus Liebig University Giessen, Schubertstrasse 81, 35392 Giessen, Germany

**Keywords:** *Caenorhabditis elegans*, mitochondria, polyphenols, aging, phytochemicals

## Abstract

The study of aging is an important topic in contemporary research. Considering the demographic changes and the resulting shifts towards an older population, it is of great interest to preserve youthful physiology in old age. For this endeavor, it is necessary to choose an appropriate model. One such model is the nematode *Caenorhabditis elegans* (*C. elegans*), which has a long tradition in aging research. In this review article, we explore the advantages of using the nematode model in aging research, focusing on bioenergetics and the study of secondary plant metabolites that have interesting implications during this process. In the first section, we review the situation of aging research today. Conventional theories and hypotheses about the ongoing aging process will be presented and briefly explained. The second section focuses on the nematode *C. elegans* and its utility in aging and nutrition research. Two useful genome editing methods for monitoring genetic interactions (RNAi and CRISPR/Cas9) are presented. Due to the mitochondria’s influence on aging, we also introduce the possibility of observing bioenergetics and respiratory phenomena in *C. elegans*. We then report on mitochondrial conservation between vertebrates and invertebrates. Here, we explain why the nematode is a suitable model for the study of mitochondrial aging. In the fourth section, we focus on phytochemicals and their applications in contemporary nutritional science, with an emphasis on aging research. As an emerging field of science, we conclude this review in the fifth section with several studies focusing on mitochondrial research and the effects of phytochemicals such as polyphenols. In summary, the nematode *C. elegans* is a suitable model for aging research that incorporates the mitochondrial theory of aging. Its living conditions in the laboratory are optimal for feeding studies, thus enabling bioenergetics to be observed during the aging process.

## 1. Aging in the Focus of the 21st Century

Over 90% of the available online research literature about aging was published after the year 2000. One of the major targets of today’s research is an understanding of this multifactorial biological process. Aging starts at birth and ultimately ends with the death of the individual [1]. In the medical sense, the process of aging only becomes relevant in later life. Above all, aging plays a major role as a risk factor for age-related diseases such as cancer, atherosclerosis, and Alzheimer’s disease [2]. Age-associated phenomena are observed throughout this process, including a reduction in cell quantity [3] and deterioration of tissue proteins, which may lead to tissue atrophy [4]. In addition, aging leads to a decrease in the metabolic rate [5], an increase in diseases, and a loss of adaptability [6,7]. These manifestations differ between individuals and depend on the organ affected [8]. The aging process is a degenerative change in the body that is associated with biological markers and can be determined by environmental factors [9]. Environmental factors related to lifestyles, such as stress, work, smoking, sun exposure, inadequate diet, physical inactivity, or few social contacts, can either speed up or slow down the aging process [10]. In today’s research, the goal is to stop this acceleration by finding countermeasures to these damaging environmental factors and thereby delay biological aging.

For the first time in history, people have a life expectancy that significantly exceeds the age of 60 [11]. In low- and middle-income countries, this increase in life expectancy is the result of a large reduction in mortality at younger ages, such as mortality during infancy and childbirth. Better therapy and treatment for infectious diseases also play a major role in these countries [12]. In countries with higher incomes, on the other hand, the increase in life expectancy is explained by a decline in mortality among the older population [13]. These extra years of life, combined with a demographic shift towards an aging population, have profound implications for each person and society as a whole. Thus, these changes offer unprecedented opportunities, pose major problems, and have a fundamental impact on current lifestyles. However, knowing that demographic changes will not stop, these changes are predictable and can be partly planned and prevented. In addition to age-related phenomena at the organ level, cellular processes are of great importance in advancing age.

Over the years, several molecular theories of aging have been put forward. These theories include the “telomere hypothesis of cell senescence”, which states that sufficient telomere loss on one or more chromosomes in normal body cells triggers cell senescence, while reactivation of the enzyme is necessary for immortality [14,15]. Other theories refer to more specific brain aging. The “calcium hypothesis of Alzheimer’s disease and brain aging” sheds light on the neuronal calcium ion, its regulation, its subcellular concentration, and its relationship to aging and genetics [16]. Over time, stress-related theories have also emerged. For example, the “metabolic stability theory of aging” states that metabolic stability, i.e., the ability of cellular regulatory processes to maintain metabolic homeostasis under stressful conditions, is the main cause of aging [17]. Nevertheless, the mitochondrial and radical-associated theories of aging are considered standard in contemporary aging research (e.g., the “free radical theory of aging” and the “mitochondrial free radical theory of aging” by Denham Harman [18]). In this article, we focus on mitochondrial theories and the possibility to use the established model organism *C. elegans* for such research.

Mitochondrial dysfunctions have long been associated with aging and age-related diseases [19]. With advancing age, mitochondrial dynamics, biogenesis, and oxidative phosphorylation capacity steadily decline [20]. In addition, damage to mitochondrial DNA (mtDNA), production of reactive oxygen species, induction of apoptotic processes, and oxidation of multiple mitochondrial proteins lead to the formation of damaging protein fragments [21].

## 2. *Caenorhabditis elegans* as a Model of Aging

The nematode *Caenorhabditis elegans* (*C. elegans*) has become one of the main model organisms for fundamental studies within molecular and cellular biology, especially in the field of aging research and genetics [22]. Sydney Brenner was one of the first to conduct research on the development of *C. elegans* as a model organism in the 1960s [23]. With the sequencing of the complete genome in 1998, *C. elegans* became accessible for molecular analyses. Of the ∼20,000 genes in *C. elegans*, an estimated 15–30% are essential, but many genes have not yet been identified or characterized [24]. It became clear that the similarities are remarkable between nematodes and humans in terms of genetic background. About 40% of the genes associated with human diseases are phylogenetically identical to those in *C. elegans*, so-called homologs [25]. This fact makes the nematode a suitable model for understanding disease development in humans. *C. elegans* offers numerous advantages as a model for the study of eukaryotes, including its small body size, high reproductive rate, ease of cultivation, low cost, long-term cryopreservation, rapid generation time, transparency, invariant cell number, and development. In addition, there is the possibility to employ gene knockdown using RNAi, a technology developed using *C. elegans* [26]. This technique offers several advantages for research with *C. elegans*. It is specific, effective, rapid, and easy to perform with the nematode model. As it results in loss-of-function phenotypes, this method is a useful tool for studying genetic interactions [27,28,29]. Over the years, several protocols have been developed to induce genetic knockdown in *C. elegans*. This knockdown can be achieved via microinjection [30], feeding [31], or soaking [32] the nematodes in dsRNA. A more recent method for genome editing is the clustered regularly interspaced short palindromic repeats (CRISPR)/Cas9 system [33]. One advantage of this tool is its ability to produce targeted mutations in the endogenous genes of the individual [34]. In addition, it is possible to incorporate fluorescent proteins into the nematode using this method to closely monitor the expression and localization of genetic products [35]. Therefore, this method is faster and less laborious than other more conventional methods of transgenesis or bombardment with microparticles to generate transgenic strains or mutants [36].

Several strains of nematode were created to screen aging in invertebrates. Other strains of nematode were created to improve research with this organism. Nematode egg laying begins from the young-adult stage, and a hermaphrodite can lay up to 300 eggs. After the sperm is used up, an increase in the number of offspring up to 1000 can be achieved by mating with males [37]. Handling this high amount of progeny requires time and patience. Therefore, sterilization of the parental generation is an established tool to avoid distorting the results. One of these methods is chemical sterilization with the DNA synthesis inhibitor 5-fluoro-2′-deoxyuridine (FUdR). By inhibiting cell division, the larvae in the eggs are prevented from further development. The eggs remain in the body of the parent animal and do not hatch [38]. Unfortunately, the use of FUdR has negative effects on mitochondria, which have a significant impact on the determination of lifespan [39]. To avoid this influence and prevent bias through a high amount of progeny, other genetic methods were used. A good example of such a method is the nematode strain PX627 created using CRISPR/Cas9 [40]. Exposure of the nematodes to auxin, a non-native, non-toxic, and cost-effective hormone, leads to the inducement of sterility in hermaphrodites and males. Recent studies showed that this auxin-inducible strain offers a far better option to maintain the physiological conditions of the mitochondria after sterilization [41].

A major advantage in research with *C. elegans* is the investigation of aging processes. As lifespan is a genetically regulated trait, several long- and short-lived mutants have been isolated. Various genes and signaling pathways have subsequently been described as regulative for longevity. A prominent candidate is the long-lived *daf-2* mutant. The insulin/insulin-like growth factor 1 receptor (I/IGF-1R) homolog DAF-2 signals through a conserved PI3 kinase/Akt pathway and ultimately inhibits the activity of DAF-16, a FOXO family transcription factor [42]. Another mutant strain that exhibits significant lifespan extension is the *age-1* mutant [43,44]. Like *daf-2*, which also depends on DAF-16, *age-1* is integrated into the PI3K/Akt pathway and encodes a homolog of mammalian PI3K [45]. One of these nematode strains is called TJ401. This *age-1* mutant strain showed a 65% increased mean lifespan compared to wild-type nematodes. Mutations, such as those in the *daf-2* and *age-1* strains, resulted in arrested larvae and forced larvae into the Dauer stage. This process increased nematode longevity and improved stress resistance [46]. The hyperresistance of these strains supports the free radical theory, one of the most important theories proposed by Denham Harman in 1956, since reactive oxygen species can modulate the transfer of DAF-16 into the nucleus, where it exerts its activity as a transcription factor [47,48]. Short-lived animals, such as *C. elegans*, generally have several advantages. Due to their short generation time and short, medium, and maximum lifespans compared to higher animals, results can be obtained within a very short period of about two weeks.

Recent studies have found that nematode longevity can be increased through feeding with polyphenols and plant extracts containing high levels of various phenolic compounds [49]. Pre-metabolized compounds also showed similar effects, although they represent metabolization by bacteria in the human gut [50]. Lifespan extension is not the only important effect of these metabolites. The use of harmful substances such as pesticides leads to a strong reduction in the vitality and lifespan of nematodes [51,52]. Simultaneous treatment with phenolic compounds can restore the vitality of these nematodes [53], possibly yielding antioxidant mechanisms and a distinct influence on life-prolonging signaling pathways such as the insulin/IGF-1 signaling pathway. The translocation of DAF-16/FOXO into the nucleus increases the expression of several life-prolonging genes, which could be responsible for the observed results [54].

## 3. Mitochondria, Aging, and *C. elegans*

As an extension of the “free radical theory of aging”, also by Denham Harman, mitochondria were added in the 1970s, leading to the “mitochondrial free radical theory of aging” [18]. The core message of this theory is that the conservative cause of mitochondrial dysfunction and, therefore, also that of aging processes in eukaryotic organisms is mitochondrial reactive oxygen species (ROS), damaging biological macromolecules. This connection between ROS, molecular damage, malfunctioning of the mitochondria, and the aging process has been explored and verified by a large number of studies to date [55,56]. When considering aging, a decrease in mitochondrial integrity and function is usually manifested. In addition, reduced ATP synthesis increased ROS production and led to lowered redox defenses. However, it is unclear to what extent these processes are natural causes or consequences of the aging process [57,58]. The relationship between mitochondria and the aging process has been described in several studies over the years. It is now believed that functional and dynamic changes in the mitochondria trigger mitochondrial dysfunction and thus contribute to aging.

Mitochondrial biogenesis is one of these changes. This mechanism is necessary for enlargement of the mitochondria by increasing their mass and number and is controlled by peroxisome proliferator-activated receptor-γ coactivator 1α (PGC-1α) [59]. It has been noted that mitochondrial biogenesis may decrease due to an age-dependent decrease in PGC-1α levels [60]. Although nematodes lack this protein, skn-1 also drives mitogenesis and thus has comparable functions to PGC-1α [61]. Regulation of mitochondrial biogenesis occurs through mitophagy, a selective form of autophagy [62]. In *C. elegans*, this process regulates mitochondrial content and nematode longevity. One of the key mediators of mitophagy and ensuring longevity under stress conditions, which is transduced to SKN-1 signaling, is DCT-1. *dct-1* expression localized to the outer mitochondrial membrane is controlled in part by the FOXO transcription factor DAF-16 and is increased in the presence of low insulin/IGF-1 signaling [63]. It has been extensively observed that mitophagy mediated by *dct-1* is involved in the aging of *C. elegans*. With advancing age, mitochondria accumulate in wild-type nematodes, leading to a deficiency of *dct-1*. The autophagy key gene *bec-1* was found to further reflect the effects of aging on mitochondrial masses in young adult animals. The induction of mitophagy was observed in long-lived *daf-2* mutants. Impairment of mitophagy by knocking down *dct-1*, *pink-1*, and *pdr-1* (the nematode Parkin homolog) significantly shortened the lifespan of *daf-2* mutants. Indeed, *dct-1* was transcriptionally induced under the control of *skn-1* and *daf-16* to remove dysfunctional mitochondria via mitophagy and coordinate mitochondrial biogenesis and mitophagy [63]. Mitophagy and mitochondrial biogenesis may work together to counteract the aging process [64]. Regarding bioenergetics as another conductor of aging, changes in basal and ATP-linked oxygen consumption rate (OCR) at the critical third larval stage were found to be a potential predictor of lifespan extension in response to mitochondrial stress by RNAi. These changes most likely precede processes of reprogramming in favor of longevity. Alterations in basal and ATP-linked OCR likely promote metabolic, genetic, and epigenetic reprogramming later in life, which are causally involved in longevity [65,66]. Consequently, mild mitochondrial stress results in less profound changes in mitochondrial functional parameters [67]. Another mechanism associated with the aging process is mitochondrial translation. This process consists of four phases: Ribosome initiation, elongation, termination, and recycling [68]. Mitochondrial translation is carried out by mammalian mitochondrial ribosomes (mitoribosomes), whose major task is to synthesize proteins essential for ATP production via oxidative phosphorylation. This mitochondrial translation is more similar to prokaryotic translation and differs from that of cytoplasmatic ribosomes [69] because mitochondrial protein synthesis requires several mitochondrial factors at each stage [70]. The main regulatory factor for the initiation of the translation process MTIF2 was also shown to play a role in pathological myocardial hypertrophy during aging and obesity [71]. Additionally, the lack of MTIF2 in Saccharomyces cerevisiae results in impaired mitochondrial protein synthesis, affecting respiration [72]. Recent studies have shown that the disruption of mitochondrial network homeostasis by blocking fusion or fission, combined with reduced mitochondrial translation, prolongs lifespan and stimulates the stress response [73]. The underlying reason for this increased lifespan is the influence of primary lysosome biogenesis and autophagy transcription factor HLH-30/transcription factor EB (TFEB) [74,75]. This result mainly suggests that mitochondrial dynamics occur downstream of mitochondrial translational stress to affect longevity and that mitochondrial dysfunction transmits stress signals to lysosomes. These processes stimulate lysosome biogenesis and, consequently, promote longevity [73]. The mitochondrial network is maintained by mitochondrial fission and fusion. These processes coordinate a mitochondrial structure that is flexible and adaptable to the changing cellular environment.

Studies in recent years have shown that mild mitochondrial dysfunction can delay aging and age-related loss of function in various animal models, including mice, *Drosophila melanogaster*, and *Caenorhabditis elegans* [76,77,78]. In addition to the mitochondrial function of mitochondria, the shape of the mitochondria is also affected by fission and fusion under these conditions. Notably, for *C. elegans*, several orthologs for mitochondrial dynamics have been found. There are two orthologs for fusion proteins (FZO-1 and EAT-3) and three for fission proteins (DRP-1, FIS-1, and FIS-2) [79]. Beyond having a clear impact on aging, the role of mitochondrial dynamics in regulating lifespan is not well understood. Regarding studies of *D. melanogaster* and *C. elegans*, mitochondrial fusion is associated with increased longevity, and age correlates with fragmentation of the mitochondrial network [80,81,82]. Mitochondrial dynamics are required for lifespan extension under various conditions of longevity, including the target of rapamycin kinase complex 1 (TORC1)-mediated longevity, AMP-activated protein kinase (AMPK)-mediated longevity, and in the presence of nutritional restriction [83,84]. A classic example of a nutrient sensor associated with longevity is TORC1. This highly conserved protein complex promotes processes such as protein translation to provide macromolecules for growth and proliferation. At the same time, this protein inhibits catabolic activities such as autophagy. Suppression of TORC1 at the genetic and pharmacological levels by rapamycin administration promotes longevity in a variety of animal species. In contrast to TORC1, the conserved kinase AMPK is activated under low-energy conditions. Activation promotes catabolic processes that generate ATP, including the TCA cycle, fatty acid oxidation, and autophagy, leading to prolonged lifespan in *C. elegans* [84].

Since mitochondrial dysfunction is one of the hallmarks of aging [57], the mitochondrial response to unfolded proteins (mtUPR) is the first response that leads to protection from stress [85]. The main role of this process is to repair or eliminate misfolded proteins to mitigate damage [86]. The underlying reaction pathway is thought to have complex effects on longevity. In *C. elegans*, this response is controlled by activating transcription factor associated with stress-1 (ATFS-1). Under stress-free conditions, ATFS-1 is degraded in the mitochondria after being imported by Lon protease [87]. Under mitochondrial stress conditions, the transfer of ATFS-1 into the mitochondria is prevented. ATFS-1 can thus enter the nucleus, where it upregulates the expression of mitochondrial chaperones, various detoxification enzymes, and metabolic enzymes [88]. An activator of mtUPR, as well as FOXO signaling, is the NAD^+^/sirtuin pathway [89]. NAD^+^ represents an important cofactor for several processes, which include the regulation of metabolic homeostasis and its function as a substrate for sirtuin deacetylases [90,91]. In *C. elegans*, the homolog of the mammalian sirtuin is *sir-2.1*, which controls mitochondrial function by deacetylating the FOXO homolog DAF-16 [92]. In a recent study, NAD^+^ precursors were shown to lead to an improvement in mitochondrial homeostasis [89] through the activation of *sir-2.1*, which led to an improvement in the disturbed balance between OXPHOS subunits encoded by mitochondrial DNA and nuclear DNA. This phenomenon is related to the activation of UPRmt, which promotes longevity, and the subsequent translocation and activation of the FOXO transcription factor *daf-16*, triggering an antioxidant protective mechanism.

In mitochondrial research, *C. elegans* offers strong and unique advantages. The degree of conservation of mitochondrial proteins between nematodes and mammals is very high, indicating that information acquisition from nematode research on mitochondria can be transferred to mammals [93]. The possibility of staining mitochondria and tracing their mobility, structure, and even function in transparent nematodes reinforces *C. elegans* as a suitable model for mitochondrial research. Assays used to study human mitochondria are easily transferable to the same studies on nematodes with minor modifications. This similarity applies to studies on oxidative phosphorylation, electron transport chain (ETC) enzyme assays, blue native gels (BNGs), free radical production, etc. [94,95]. It should be noted, however, that isolation of functional mitochondria from nematodes is somewhat more difficult than that from mammalian cells, primarily because the cuticle must be disrupted while the intact mitochondria remain intact. A useful tool for this process is the Balch homogenizer. This device allows rapid and careful homogenization and permeabilization of the nematode and the extraction of functional organelles [96]. A very simple method for this process is to create nuclear defects in the mitochondria using RNAi. Thus, it is possible to generate different degrees of mitochondrial damage via RNAi treatment and thereby investigate many mitochondrial proteins whose knockouts can be lethal for the animal [97].

To date, several mutant nuclear-encoded and mitochondrial-encoded subunits of the respiratory chain complexes have been studied in *C. elegans*. In addition, mutations that lead to an inhibition of the synthesis of coenzyme Q have been identified [98]. These, as well as proteins that indirectly modify the complexes of the respiratory chain, have been investigated at the molecular and cellular level, respectively, and in the whole animal model through a range of experimental approaches. In these models, the dysfunction of individual respiratory chain complexes can lead to increased or decreased lifespan [99,100], neuromuscular deficits [101], restricted development [102], reduced fertility [103], or altered anesthetic sensitivity [104]. All of these symptoms mimic those that occur in people with mitochondrial damage. Furthermore, the dysfunction of mitochondrial respiratory chain complexes can affect gene expression profiles [105,106], as well as ROS formation [107,108]. Furthermore, mutants of mitochondrial respiratory chain complexes with an altered life span are useful in studying the contribution of energy consumption, ROS, and stress responses to the process of aging.

Nevertheless, there are major differences between mitochondria in mammals and those in *C. elegans*. The approximately 14 kb circular mitochondrial DNA (mtDNA) of *C. elegans* contains homologs of 36 of the 37 genes found in humans. However, the ATP8 subunit of complex V is missing in *C. elegans* [109]. In addition, *C. elegans* appears to have fewer copies of mtDNA per cell than humans [110,111]. Instead of coenzyme Q10, which has a chain of ten isoprenyl repeats, *C. elegans*, like rodents, mainly harbors coenzyme Q9 [112]. While the glyoxylate cycle is not normally found in animals, *C. elegans* has a malate synthase/isocitrate lyase that cleaves isocitrate to glyoxylate and succinate [113]. In contrast to plants, this metabolic pathway is encoded by a gene in nematodes, which gives the nematodes the ability to divide the TCA cycle and thus form fragments with two carbon atoms. These atoms are then used for anabolic processes. Furthermore, isolated intact *C. elegans* mitochondria can respire using malate as a substrate. However, malate is a poor substrate for mammalian mitochondria. In addition, mitochondria from *C. elegans* are less sensitive to some artificial uncouplers of the respiratory chain, as well as to theonoyltrifluoroacetone (TTFA), an inhibitor of complex II in mammals [114].

Despite these differences, mitochondria originating from *C. elegans* are very similar to those of mammals. Thus, research on mitochondria in the *C. elegans* model organism provides a useful system for investigating unanswered questions about mitochondrial bioenergetics and dysfunction.

## 4. Polyphenols and Secondary Plant Metabolites in Aging Research

The effects of plants or plant extracts from traditional Indian and Chinese medicine have been known for centuries [115]. Today, these effects can be traced back to their individual components, so-called natural substances. These substances represent a large family of substances with a broad spectrum of biological activity. Natural substances are mainly obtained from plants but also from bacteria, fungi, and marine sources. The physiologist and Nobel Prize winner Albrecht Kossel divided them into primary (e.g., proteins, lipids, and carbohydrates) and secondary substances (e.g., pheromones, alkaloids, phenols, and steroids) [116]. Such a classification is now outdated but is still used in the literature for historical reasons. Today, it is known that a natural substance can have life-sustaining functions, as well as the functions of a classical secondary metabolism.

More than 200,000 plant compounds are currently known and are divided into several groups depending on their chemical properties [117]. Phenolic compounds form the largest group, with more than 8000 different polyphenols known [118]. Depending on their structure, the group of polyphenols can be divided into 10 subgroups. The most important and largest group is the group of flavonoids, which comprises about 4000 known compounds [119].

Due to their ubiquitous occurrence in many fruits and vegetables, flavonoids are a daily component of the human diet [120]. High polyphenol contents are found, for example, in grapes, pomegranates, apples, and various types of tea. Of particular note are grapes and apples, which can have a polyphenol content of approximately 200–300 mg polyphenols per 100 g fresh weight [121]. The pleiotropic effects of polyphenols include anti-allergenic, anti-inflammatory, anti-microbial, anti-oxidative, and anti-aging effects. In addition, polyphenols have protective effects on blood vessels and the heart and can prevent thrombosis [122]. Anti-cancer and neuroprotective effects have also been described [123,124,125] (Table 1).

*C. elegans* provides a solid model for studying the effects of polyphenols on the aging system. The administration of substances to the nematode’s natural diet is a suitable way to observe the effects of the compound on the animal [177]. Starting in 1974, the first drug tests were performed on *C. elegans* by incorporating substances into the agar of plates [23]. In contrast to the current use of substances, this method consumed large amounts of substances and was labor intensive. Although this type of method was maintained for several observations, the nematode eventually evolved into a suitable model for high-throughput screening (HTS) [178]. The main advantages of the nematode HTS model are the availability of proven genetic tools and genomic resources (RNAi and CRISPR/Cas9) [33,179]. The complexity of the whole organism system improves the chances of identifying agents that will ultimately be more effective in more complex organisms such as humans [180]. The presence of a large number of phenotypes opens the possibility to observe visual changes after drug administration [181]. The ability to study absorption, distribution, metabolism, excretion, and toxicity (ADMET) and drug efficacy, as well as the ability to model complex human diseases that are not easily reproduced in other in vitro models, affords various pharmaceutical and medical opportunities [182]. The use of automated microscopic devices, microplate readers, and automated worm transfer has simplified and accelerated the screening of large quantities of bioactive compounds [183,184].

## 5. Mitochondria as a Target of Phytochemicals

Before polyphenols can act as bioactive agents, they must reach the cells or compartments intended for them. Polyphenols undergo various biotransformations, not only through digestive enzymes but also (especially) by the microbiota, which changes the chemical structure and properties of the molecules consumed [185]. The human intestinal microbiota consists of approximately 10^12^–10^14^ bacterial cells and is an extremely diverse entity involved in the digestion and fermentation of food components such as polyphenols. In this context, the microbiota interacts closely with the immune system, making a balanced microbiota essential for maintaining a healthy state [186,187]. Ultimately, traceability in plasma reveals the availability of various phenolic acids after microbial remodeling. Therefore, it is important to note that the phenolic compounds circulating in the human system can differ greatly from those administered. Only those that reach the desired tissue can exhibit their bioactive functions [188] (Figure 1). The best known and most extensively described are the antioxidant properties of polyphenols, which give them the ability to scavenge ROS [189,190]. Since ROS play a crucial role in the development of various diseases [191,192], antioxidants, which play a protective role against free radicals, are thought to be beneficial in preventing these diseases [193]. The antioxidant properties of polyphenols are also thought to protect against inflammation. In this respect, such properties could also be useful in inflammation-related diseases such as autoimmune diseases [194,195]. Although polyphenols can directly scavenge ROS, new evidence suggests that consumable amounts of polyphenols are not sufficient for this outcome. Rather, it is assumed that polyphenols activate, among others, the Keap1/Nrf2/ARE signaling pathway, which triggers a hormonal activation of phase II enzymes and thus strengthens the body’s oxidative defense system [196,197]. In mammals, the *Drosophila melanogaster* and *C. elegans* detoxification pathways are tightly regulated, making their basic activity low. Contact with toxic xenobiotics or other oxidants then simultaneously activates the expression of several genes through inducible transcription factors [198]. SKN-1 is a transcription factor orthologous to the mammalian Nrf2. Activated by various xenobiotics, oxidants, and electrophiles, SKN-1 confers resistance by activating detoxification genes [199,200,201,202].

In addition to direct antioxidant mechanisms, polyphenols have also been shown to initiate indirect mechanisms that promote innate detoxification pathways. One of these mechanisms is hormesis [203]. Hormesis refers to a biphasic, dose-dependent effect of bioactive substances. While high concentrations are considered toxic, moderate to low doses of exposure can be beneficial to health and activate cellular adaptive mechanisms [204]. A very prominent hermetic process in aging research is caloric restriction. Reduced caloric intake was shown to increase life expectancy in subjects [205]. In addition, it was reported that phytochemicals can be neuroprotective by exhibiting hermetic processes through the involvement of various genes [196]. Heat shock proteins should also be mentioned because temperature is an important hermetic factor. The induction of heat shock leads to the upregulation of heat shock proteins and chaperones. These types of proteins preserve the three-dimensional structures of proteins and help newly synthesized proteins fold correctly [206,207]. In *C. elegans*, variation in hormesis effects was shown to be genetically determined. These results confirm that hormesis is formed by mechanisms that were optimized during evolution [208]. A recent study in *C. elegans* showed that the phenolic acids protocatechuic acid, gallic acid, and vanillic acid trigger the hormesis process [50].

Besides the ability of polyphenols to scavenge ROS, other mechanisms in mitochondria have been described and studied. These mechanisms include the complex activity of the ETC, which was recently improved by the phenolic compound protocatechuic acid [76]. Energy release in the form of ATP was also described as being altered after treatment with various polyphenols [209,210,211]. In addition, the influence of phenolic compounds and metabolites on mtDNA was observed [49]. Resveratrol is a polyphenol found in grapes and red wine that possesses several biological activities, including anti-inflammatory and antioxidant activities [212]. Recent studies reported an increase in mtDNA quantity after treatment with resveratrol [213,214]. Thus, resveratrol may lead to an elevated level of mitochondrial biogenesis [215]. Other effects of polyphenols on mtDNA have also been demonstrated in disease models. As with resveratrol, the incubation of MDA-MB-231 human breast cancer cells with this compound results in decreased levels of mtDNA [216]. This result may be due to increased autophagy induced in response to mtDNA damage. Similarly, treatment with curcumin in HepG2 human hepatoma cells leads to increased damage to mtDNA [217]. This damage was shown to trigger apoptosis in these cancer cell lines. Another effect of resveratrol on aging in *C. elegans* was demonstrated through direct interaction with mitochondrial respiration. By decreasing the activity of mitochondrial respiration, resveratrol prolongs lifespan through a mechanism related to caloric restriction. In this case, *sir-2.1* was increased after the treatment of nematodes with resveratrol [218]. Influences on mitophagy were also observed. Catechinic acid resulted in a prolonged lifespan and a reduction in age-related behaviors by regulating genes associated with mitophagy pathways in *C. elegans*. By affecting *bec-1* and *pink-1*, mitochondrial phagocytosis was induced at early stages and lifespan may be affected [219]. The mechanism of mitophagy leads to the elimination of accumulated dysfunctional mitochondria, which can prolong lifespan. Epigallocatechin-3-gallate was shown in recent studies to be able to restore mitochondrial function by increasing biogenesis in nematodes, thereby improving the redox status of nematodes [220]. Another interesting polyphenol that showed direct influences on the respiratory chain is oenothein B. This hydrolyzable polyphenol from the tannin group showed health-promoting effects via *isp-1*, including a reduction of ROS accumulation, improvement of motility flexibility, and aging pigments. The gene *isp-1* encodes a subunit of the mitochondrial complex III in *C. elegans*, also known as the coenzyme Q–cytochrome c oxidoreductase [221].

## 6. Conclusions

In summary, the *C. elegans* nematode model offers great advantages for mitochondrial research, especially by elucidating aging phenomena in a nutritional and environmental context. Due to the ease of handling this model organism, the high rate of mitochondrial maintenance, and the close monitoring of food sources, this animal is an ideal candidate for monitoring much more than the aging process itself. Future nutritional scientists working with phenolic compounds and extracts from fruits and vegetables should consider *C. elegans* as a suitable model for their research purposes. Understanding interspecies transferability will also play a major role in future research to find alternative models to classical animal models for drug or natural substance research. Unveiling new effects of secondary plant compounds on metabolic pathways will be a key task for researchers going forward. In addition, the potential application of these compounds as therapeutics will be of great interest. If the aging process cannot be halted, youthful physiology could potentially be maintained into old age. The pharmaceutical treatment of age-related diseases could also be supported with natural substances, and the time of illness could be shortened.

## Figures and Tables

**Figure 1 biomolecules-12-01550-f001:**
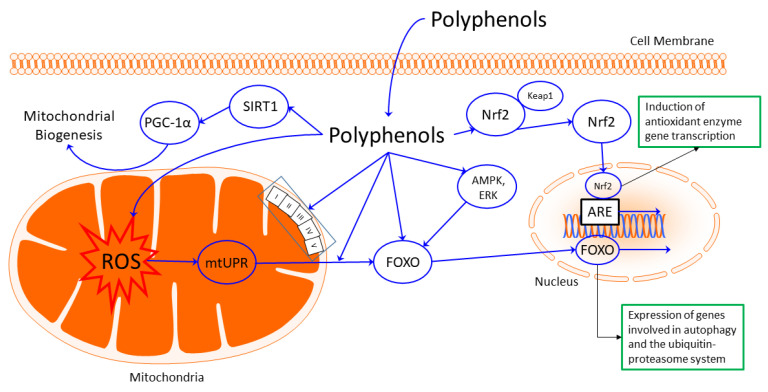
Possible interactions between polyphenols and metabolic pathways associated with aging. After biotransformation, polyphenols may acquire the ability to cross the cell membrane into the cytosol. It is in the cytosol that the polyphenols exert their effects. Mitochondrial biogenesis is stimulated by polyphenols via an SIRT1-activated PGC-1α-mediated mechanism. Mitochondrial stress triggered by ROS production in the mitochondria can be averted either directly or indirectly by ROS scavenging or by the influence of polyphenols on FOXO transcription factor activation. Polyphenols cause the dissociation of Keap1 from the Nrf2/Keap1 complex. Translocation of Nrf2 to the nucleus leads to its association with ARE in the regulatory regions of target genes. This process induces the transcription of antioxidant and detoxification enzymes. Some polyphenols can directly affect oxidative phosphorylation or complexes of ETC. In this way, energy performance can be changed.

**Table 1 biomolecules-12-01550-t001:** Classes of polyphenols with their commonly used agents, studied effects, and corresponding important literature.

Classes of Polyphenols	Commonly Used Representatives	Studied Effects	Literature
Flavone	Luteolin	Sensitizes cancer cells to therapeutically induced cytotoxicity by suppressing cell survival pathways (PI3K/Akt, NF-κB, and XIAB) Stimulates apoptosis pathways inducing the tumor suppressor p53	[126,127,128,129,130]
	Baicalein	Anti-malignant potential through the influence of several signaling cascades (MAPK, mTOR, PKB/Akt, PARP, MMP-2, MMP-9, and caspase)	[131,132,133,134,135]
	Apigenin	Induces intrinsic apoptosis pathways Leads to the downregulation of matrix metallopeptidases Leads to downregulation of PI3K/Akt/NF-κB signaling	[136,137,138,139]
Flavonol	Kaempferol	Induces apoptosis and cell cycle arrest at the G2/M phase Downregulation of the epithelial–mesenchymal transition (EMT)-related markers, and PI3K/PKB signaling pathways	[140,141,142,143]
	Myricetin	Therapeutic effects on atherosclerosis, thrombosis, diabetes, and Alzheimer’s disease Regulates MAPK, PI3K,/Akt/mTOR, IκB/NF-κB, and AChE Enhances immunomodulatory functions	[144,145,146,147,148]
	Quercetin	Inhibits NF-κB Activates SIRT1 by improving the NAD^+^ level Inhibits α-glucosidase and increases adiponectin Decreases the activity of inflammatory enzymes such as 5-LOX, 12-LOX, COX, NOS, and MPO	[149,150,151,152]
Flavanone	Hesperitin	Improves mitochondrial function by increasing complex function Upregulates antioxidant levels (SOD, GPx, and GR) by triggering PI3K/Akt pathway Offers neuroprotection by regulating the TLR4/NF-κB signaling pathway Augments antioxidant cellular defenses via the ERK/Nrf2 signaling pathway	[153,154,155,156]
	Naringenin	Inhibits TNF-α-induced TLR2 expression by inhibiting activation of the NF-κB and c-Jun NH2-terminal kinase pathways Modulates the MAPK signaling pathway Offers protective effects against LPS-induced injury	[157,158,159]
Flavanonol	Taxifolin	Offers anti-inflammatory effects through suppressing NF-κB activation Offers hepatoprotective effects through reduced CD4^+^ and CD8^+^ T cells in injured liver tissue Downregulates the levels of TNF-α and COX-2	[160,161,162]
	Engeletin	Mitigates Aβ1-42-induced oxidative stress and neuroinflammation through the Keap1/Nrf2 pathway Offers hepatoprotective effects through activation of PPAR-γ Reduces NF-κB-dependent signaling	[163,164,165]
Anthocyanidin	Malvidin	Inhibits IL-6, TNF-α, and IL-1β Increases antioxidative enzymes (SOD and GPx) Stimulates AMPK-mediated autophagy	[166,167]
	Delphinidin	Increases expression of antioxidant protein Nrf2-related phase II enzyme heme oxygenase-1 (HO-1) Modulates JAK/STAT3 and MAPKinase signaling to induce apoptosis	[168,169]
Flavan-3-ol	Epigallocatechin gallate	Offers anti-viral and anti-bacterial effects Inhibits tumor necrosis factor-α (TNF-α)-induced production of monocyte chemoattractant protein-1 (MCP-1)	[170,171]
Isoflavone	Daidzein	Blocks the transcriptional activation of pro-inflammatory genes and decreases the mRNA level of Cxcl2 in TNF-α-treated cells Increases AMPK phosphorylation followed by GLUT 4 translocation and glucose uptake	[172,173]
	Genistein	Induces apoptosis through activation of caspase-1 Offers anti-proliferative effects through downregulation of DNA methylation Suppresses Akt activity, promoting deactivation of NF-κB	[174,175,176]

## Data Availability

Not applicable.

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
