# Peer review of "Caenorhabditis elegans as a Model for the Effects of Phytochemicals on Mitochondria and Aging"

_biomolecules, 2022, doi:10.3390/biom12111550_

Round 1

Reviewer 1 Report

In the present manuscript, the authors provided an overview of C. elegans as an aging research animal model, the signaling pathways and biological processes involved in aging, the biology of mitochondria, and the effect of phytochemicals in mitochondria-deponent and -independent aging. They also described the advantages of using C. elegans in aging and mitochondrial research.

It is not apparent from the abstract what the reader is expected to learn from the review article, and the authors should modify the abstract to reflect the key expected learnings from this paper.

Line 25: “Beginning with birth aging starts and ultimately 25 ends in the death of the individual” I could not find data in Reference #1 to support this claim. A more appropriate primary source must be cited.

Line 83: Besides RNAi, authors should include CRISPR as a tool for genetics manipulations.

C. elegans is an excellent model for high throughput screening studies. Authors should include the potential of C. elegans in high throughput chemical screens in the context of phytochemicals.

Line 84: “Due to their short generation time and short, medium, and maximum life span, results can be generated within a very short period of about two weeks.” 

Here, the authors suggest that the aging experiments can be completed in two weeks. Although this is true for short-lived animals, it excludes long-lived animals. Authors should make text changes to include experiments where animals (e.g., daf-2 mutants mean life span of 60+ days) are long-lived.

Line 87:” 10% of the approximately 18,600 C. elegans genes have been characterized and mapped. Many C. elegans genes are orthologues to those of higher organisms.” 

Authors should include a recent reference to support this claim.

Line 99: Authors may describe the mode of action for FUDR, e.g., it inhibits cell division, reduces egg production, and prevents eggs from hatching.

Figure 1 seems unnecessary and appears to be out of the scope of this paper. A more appropriate figure may depict a schematic representation of phytochemicals and their effect on mitochondria. How mitochondrial defects affect aging and key pathways involved in this process. 

Similar to Figure 1, the data represented in Table 1 is less engaging. Instead, a table of different classes of phytochemicals and their mode of action accompanied with relevant citations will be helpful for the reader to understand the diversity of phytochemicals and gain a broad understanding of their biological significance.

The authors need to briefly summarize the relationship between mitochondrial function and aging. Especially how mito-defects negatively affect longevity. 

Following are a few papers highlighting the importance of bioenergetic, biogenesis and turnover of mitochondria in aging.

Palikaras K elt al., Coordination of mitophagy and mitochondrial biogenesis during aging in C. elegans

Silvia Maglioni et al., Mitochondrial bioenergetic changes during development as an indicator of C. elegans health-span

Yasmine J. Liu et al., Mitochondrial translation and dynamics synergistically extend lifespan in C. elegans through HLH-30

Yue Zhang, et al., Neuronal TORC1 modulates longevity via AMPK and cell nonautonomous regulation of mitochondrial dynamics in C. elegans

Briefly summarize the role of Mitochondrial unfolded protein response and mitochondrial metabolite (NAD+) in longevity.

Authors should provide future directions and describe key questions that can be addressed using C. elegans as a model to understand the role of phytochemical research in aging.

Author Response

Reviewer 1

In the present manuscript, the authors provided an overview of C. elegans as an aging research animal model, the signaling pathways and biological processes involved in aging, the biology of mitochondria, and the effect of phytochemicals in mitochondria-deponent and -independent aging. They also described the advantages of using C. elegans in aging and mitochondrial research.

  1. It is not apparent from the abstract what the reader is expected to learn from the review article, and the authors should modify the abstract to reflect the key expected learnings from this paper.
  • We thank you for the correction of the abstract and have changed it as follows: The study of aging is an important topic in contemporary research. Considering the de-mographic change and the resulting shift towards an older population, it is of great interest to preserve youthful physiology in old age. For this endeavor, it is necessary to choose an appropriate model. One such model is the nematode Caenorhabditis elegans (C. elegans), which has a long tradition in aging research. In this review article, we focus on the advantages of using the nematode model in aging research, focusing on bioenergetics and the study of secondary plant metabolites that have interesting implications during this process. In the first section, we review the situation of aging research today. Conventional theories and hypotheses about the ongoing aging process will be presented and briefly explained. The second section focuses on the nematode C. elegans and its utility for aging and nutrition research. Two useful genome editing methods for monitoring genetic interactions (RNAi and CRISPR/Cas9) are presented. Because of the mitochondrial influence on aging, we also introduce the possibility of observing bioenergetics and respiratory phenomena in C. ele-gans. We will then report on mitochondrial conservation between vertebrates and inverte-brates. Here we will explain why the nematode is a suitable model for the study of mito-chondrial aging. In the fourth section, we focus on phytochemicals and their application in contemporary nutritional science with emphasis on aging research. As an emerging field of science, we conclude this review in the fifth section with several studies focusing on mi-tochondrial research and the effects of phytochemicals such as polyphenols. In summary, the nematode C. elegans is a suitable model for aging research that incorporates the mito-chondrial theory of aging. Its living conditions in the laboratory are optimal for feeding studies with the end result that bioenergetics can be observed during the aging process.
  • The new abstract was integrated into the manuscript (lines 8 to 29).
  1. Line 25: “Beginning with birth aging starts and ultimately 25 ends in the death of the individual” I could not find data in Reference #1 to support this claim. A more appropriate primary source must be cited.
  • Thank you for this important point. We now cited the study from of Kinzina et. al from 2019 with the title “Patterns of Aging Biomarkers, Mortality, and Damaging Mutations Illuminate the Beginning of Aging and Causes of Early-Life Mortality.” In this interesting study the authors compare the accumulation of genetic and epigenetic damage with the selection against deleterious alleles. As a result of both factors the mortality of individuals is increasing over age and leads to death.
  • The important paper has been discussed now (lines 37)
  1. Line 83: Besides RNAi, authors should include CRISPR as a tool for genetics manipulations.
  • We agree that the CRISPR system has to incorporated into the article. We intergrated the following text: In addition, there is the possibility of gene knockdown using RNAi, a technology de-veloped using C. elegans. This technique offers several advantages for research with C. elegans. It is specific, effective, rapid, and easy to perform with the nematode model. Because it results in loss-of-function phenotypes, this method is a useful tool for studying genetic interactions. Over the years, several protocols have been developed to induce genetic knockdown in C. elegans. This can be achieved by microinjection, feeding, or soaking the nematodes in dsRNA. A more recent method of genome editing is the CRISPR/Cas9 (clustered regularly interspaced short palindromic repeats) system. One advantage of this tool is the ability to make tar-geted mutations in endogenous genes of the individual. In addition, it is possible to incorporate fluorescent proteins into the nematode using this method to closely monitor the expression and localization of genetic products. Therefore, this meth-od is faster and less laborious than other, more conventional methods of transgenesis or bombardment with microparticles to generate transgenic strains or mutants.
  • This important information and the necessary references has now been incorporated into the text (lines 93 to 106)
  1. C. elegans is an excellent model for high throughput screening studies. Authors should include the potential of C. elegans in high throughput chemical screens in the context of phytochemicals.
  • To study the effects of polyphenols on the aging system, C. elegans provides a sol-id model. Administration of substances to the nematode's natural diet is a suitable way to observe the effects of the compound on the animal. Starting in 1974, the first drug tests were performed on C. elegans by incorporating substances into the agar of the plates. In contrast to the current use of substances, this method consumed large amounts of substances and was labor intensive. Although this type of method was maintained for several observations, the nematode evolved into a suitable model for high-throughput screening (HTS). The main advantages of the nematode HTS model are the availability of proven genetic tools and genomic resources (RNAi, CRISPR/Cas9). The complexity of the whole organism system improves the chances of identifying agents that will ultimately be more effective in more complex organisms such as humans. A large number of phenotypes harbors the possibility to observe visual changes after drug administration. The ability to study ADMET (absorption, distribution, metabolism, excretion, toxicity) and drug efficacy, and the ability to model complex human diseases that are not easily reproduced in other in vitro models enables pharmaceutical and medical opportunities. The use of auto-mated microscopic devices, a microplate reader, and automated worm transfer has simplified and accelerated the screening of large quantities of bioactive compounds.
  • This point and the necessary literature has also been added to the text (lines 231 to 248)
  1. Line 84: “Due to their short generation time and short, medium, and maximum life span, results can be generated within a very short period of about two weeks.” 
    Here, the authors suggest that the aging experiments can be completed in two weeks. Although this is true for short-lived animals, it excludes long-lived animals. Authors should make text changes to include experiments where animals (e.g., daf-2 mutants mean life span of 60+ days) are long-lived
  • A major advantage in research with C. elegans is the investigation of aging pro-cesses. Because lifespan is a genetically regulated trait, several long- and short-lived mutants have been isolated. Various genes and signaling pathways have then been described as regulative for longevity. A prominent candidate is the long-lived daf-2 mutant. The insulin/insulin-like growth factor 1 receptor (I/IGF-1R) homologue DAF-2 signals through a conserved PI3 kinase/Akt pathway and ultimately inhibits the activ-ity of DAF-16, a FOXO family transcription factor. Another mutant strain, which shows a significant increase in lifespan, is the age-1 mutant. Short-lived animals harbors several advantages as well. Due to their short generation time and short, medium, and maximum life span, results can be generated within a very short period of about two week.
  • This test change has been added to the discussion (lines 107 to 116)
  1. Line 87:” 10% of the approximately 18,600 C. elegans genes have been characterized and mapped. Many C. elegans genes are orthologues to those of higher organisms.” Authors should include a recent reference to support this claim.
  • We changed the claim as follows: With the sequencing of the complete genome, C. elegans is suitable for molecular anal-yses Of the ∼20,000 genes in C. elegans, it is estimated that 15-30% are essential, but many are not yet identified or characterized.
  • This point has also been added to the discussion (lines 116 to 119)
  1. Line 99: Authors may describe the mode of action for FUDR, e.g., it inhibits cell division, reduces egg production, and prevents eggs from hatching.
  • We changed the text as follows: One of these methods is chemical sterilization with the DNA synthesis inhibitor 5-fluoro-2'-deoxyuridine (FUdR). By inhibiting cell division, the larvae in the eggs are prevented from further development. The eggs remain in the body of the parent ani-mal and do not hatch. Unfortunately, the use of FUdR has negative effects on mi-tochondria, which have a significant impact on the determination of lifespan.
  • This point has also been added to the discussion (lines 130 to 134)
  1. Figure 1 seems unnecessary and appears to be out of the scope of this paper. A more appropriate figure may depict a schematic representation of phytochemicals and their effect on mitochondria. How mitochondrial defects affect aging and key pathways involved in this process.
  •  

Figure 1: Possible interactions between polyphenols and metabolic pathways associated with aging. After biotransformation, polyphenols may acquire the ability to cross the cell membrane into the cytosol. It is in the cytosol that the polyphenols exert their effects. Mitochondrial biogenesis is stimulated by polyphenols via a SIRT1-activated PGC-1α-mediated mechanism. Mitochondrial stress triggered by ROS production in mitochondria can be averted either directly or indirectly by ROS scavenging or by the influence of polyphenols on FOXO transcription factor activation. Polyphenols cause dissociation of Keap1 from the Nrf2/Keap1 complex. Translocation of Nrf2 to the nucleus leads to its association with ARE in the regulatory region of target genes. This induces the transcription of antioxidant and detoxification enzymes. Some polyphenols have the ability to directly affect oxidative phosphorylation or complexes of ETC. Therefore, the energy performance can be changed.

  • This new figure has been added to the manuscript (lines 341)
  1. Similar to Figure 1, the data represented in Table 1 is less engaging. Instead, a table of different classes of phytochemicals and their mode of action accompanied with relevant citations will be helpful for the reader to understand the diversity of phytochemicals and gain a broad understanding of their biological significance.

Classes of polyphenols

Commonly used representatives

Studied effects

Literature

Flavone

Luteolin

·       sensitizes cancer cells to therapeutically induced cytotoxicity by suppressing cell survival pathways (PI3K/Akt, NF-κB, XIAB)

·       stimulates apoptosis pathways inducing the tumor suppressor p53

[99–103]

Baicalein

·       Anti-malignant potential through influence of several signaling cascades (MAPK, mTOR, PKB/Akt, PARP, MMP-2, MMP-9, caspase)

[104–108]

Apigenin

·       Induces intrinsic apoptosis pathways

·       Leads to downregulation of matrix metallopeptidases

·       Leads to downregulation of PI3K/Akt/ NF-κB signaling

[109–112]

Flavonol

Kaempferol

·       Inducement of apoptosis, cell cycle arrest at the G2/M phase

·       Downregulation of the epithelial-mesenchymal transition (EMT)-related markers, and PI3K/PKB signaling pathways

[113–116]

Myricetin

·       Therapeutic effects on atherosclerosis, thrombosis, diabetes, Alzheimer’s disease

·       Regulates MAPK, PI3K,/Akt/mTOR, IκB/NF –κB, AChE

·       Enhances immunomodulatory functions

[117–121]

Quercetin

·       Inhibits NF –κB

·       Activates SIRT1 via improving the NAD+ level

·       Inhibit α-glucosidase and increase adiponectin

·       Decreases the activity of inflammatory enzymes such as 5-LOX, 12-LOX, COX, NOS, and MPO

[122–125]

Flavanone

Hesperitin

·       Improve mitochondrial function through increasing complex function

·       Upregulate antioxidant levels (SOD, GPx, GR)by triggering PI3K/Akt pathway

·       Neuroprotection by regulating the TLR4/NF-κB signaling pathway

·       Augment the antioxidant cellular defenses via the ERK/Nrf2 signaling pathway

[126–129]

Naringenin

·       Inhibits TNF-α-induced TLR2 expression by inhibiting the activation of NF-κB and c-Jun NH2-terminal kinase pathways

·       Modulate the MAPK signaling pathway

·       Protective effect against LPS-induced injury

[130–132]

Flavanonol

Taxifolin

·       Anti-inflammatory through suppressing NF-κB activation

·       Hepatoprotective through reduced CD4+ and CD8+ T cells in injured liver tissue

·       Down-regulating the levels of TNF-α, COX-2

[133–135]

Engeletin

·       Mitigates Aβ1-42-induced oxidative stress and neuroinflammation through the Keap1/Nrf2 pathway.

·       Hepatoprotective through activation of PPAR-γ

·       Reduce NF-κB-dependent signaling

[136–138]

Anthocyanidin

Malvidin

·       Inhibit of IL-6, TNF-α, and IL-1β

·       Increase antioxidative enzymes (SOD, GPx)

·       Stimulate AMPK-mediated autophagy

[139,140]

Delphinidin

·       Increase expression of antioxidant protein HO-1 (Nrf2-related phase II enzyme heme oxygenase-1)

·       Modulate JAK/STAT3 and MAPKinase signaling to induce apoptosis

[141,142]

Flavan-3-ol

Epigallocatchin
gallate

·       Antiviral and antibacterial effects

·       Inhibit tumor necrosis factor-α (TNF-α)-induced production of monocyte chemoattractant protein-1 (MCP-1)

[143,144]

Isoflavone

Daidzein

·       Block the transcriptional activation of pro-inflammatory genes and decreased the mRNA level of Cxcl2 in TNFα-treated cells

·       Increase the AMPK phosphorylation followed by GLUT 4 translocation and glucose uptake

[145,146]

Genistein

·       Induce apoptosis through activation of caspase-1

·       Anti-proliferative effects through down-regulation of DNA methylation

·       Suppress Akt activity, promoting deactivation of NF-κB

[147–149]

  •  
  • This new table has been added to the manuscript (line 282)
  1. The authors need to briefly summarize the relationship between mitochondrial function and aging. Especially how mito-defects negatively affect longevity.
    Following are a few papers highlighting the importance of bioenergetic, biogenesis and turnover of mitochondria in aging.
  • Palikaras K elt al., Coordination of mitophagy and mitochondrial biogenesis during aging in  elegans

  • Silvia Maglioni et al., Mitochondrial bioenergetic changes during development as an indicator of  eleganshealth-span

  • Yasmine J. Liu et al., Mitochondrial translation and dynamics synergistically extend lifespan in  elegansthrough HLH-30

  • Yue Zhang, et al., Neuronal TORC1 modulates longevity via AMPK and cell nonautonomous regulation of mitochondrial dynamics in  elegans

  • The relationship between mitochondria and the aging process has been described in several studies over the years. It is now believed that functional and dynamic changes in mitochondria trigger mitochondrial dysfunction and thus contribute to aging. Mi-tochondrial biogenesis is one of these changes. This mechanism is necessary for the enlargement of mitochondria by increasing their mass and number. Regulation of mitochondrial biogenesis occurs through mitophagy, a selective form of autophagy. In C. elegans, this process regulates mitochondrial content and nematode longevity. One of the key mediators of mitophagy and ensuring longevity under stress conditions, which is transduced to SKN-1 signaling, is DCT-1. dct-1 expression localized to the outer mi-tochondrial membrane is controlled in part by the FOXO transcription factor DAF-16 and is increased in the presence of low insulin/IGF-1 signaling. Concerning bioenergetics as another conductor of aging, it was observed that reduction of mitochondrial function, e.g., by RNAi to achieve mitochondrial stress during larval development, leads to a life-prolonging effect. Knockdown of several mitochondrial proteins significantly increased lifespan, suggesting that basal oxygen consumption rate and ATP-linked respiration positively correlate with lifespan extension. Another mecha-nism associated with the aging process is mitochondrial translation. This process consists of four phases: Ribosome initiation, elongation, termination, and recycling. Recent studies have shown that disruption of mitochondrial network homeostasis by blocking fusion or fission, combined with reduced mitochondrial translation, prolongs lifespan and stimulates the stress response. The mitochondrial network is maintained by mitochondrial fission and fusion. These processes coordinate a flexible and adaptable mitochondrial structure to the changing cellular environment. In addition, Target of Rapamycin Complex 1 (TORC1) and AMP-activated protein kinase (AMPK) have been observed to be responsible for mitochondrial dynamics such as fusion or fission. A classic example of a nutrient sensor associated with longevity is TORC1. This highly conserved protein complex promotes processes such as protein translation to provide macromolecules for growth and proliferation. At the same time, it inhibits catabolic activities such as AUtophagy. Suppression of TORC1 at the genetic and pharmacolog-ical levels by rapamycin administration promotes longevity in a variety of animal species. In contrast to TORC1, the conserved kinase AMPK is activated under low-energy conditions. Activation promotes catabolic processes that generate ATP, including the TCA cycle, fatty acid oxidation, and autophagy. This leads to a prolongation of lifespan in C. elegans..
  • This summary of the relationship between mitochondrial function and aging has also been added to the discussion (lines 156 to 188)

  1. Briefly summarize the role of Mitochondrial unfolded protein response and mitochondrial metabolite (NAD+) in longevity.
  • Since mitochondrial dysfunction is one of the hallmarks of aging, the mitochondrial response to unfolded proteins (mtUPR) is the first response that leads to protection from stress. The main role of this process is to repair or eliminate mis-folded proteins to mitigate damage. The underlying reaction pathway is thought to have complex effects on longevity. In C. elegans, this response is controlled by ATFS-1 (Activating Transcription Factor associated with Stress-1). Under stress-free conditions, ATFS-1 is degraded in mitochondria after being imported by Lon protease. Under mitochondrial stress conditions, the import of ATFS-1 into mitochondria is prevented. ATFS-1 can thus enter the nucleus, where it upregulates the expression of mitochondrial chaperones, various detoxification enzymes, and metabolic enzymes. An activator of mtUPR, as well as FOXO signaling, is the NAD+/sirtuin pathway [68]. NAD+ represents an important cofactor for several processes. These include the regulation of metabolic homeostasis and its function as a substrate for sirtuin deacetylases. In C. elegans, the homolog of the mammalian sirtuin is sir-2.1, which con-trols mitochondrial function by deacetylating the FOXO homolog DAF-16. In a re-cent study, NAD+ precursors were shown to lead to an improvement in mitochondrial homeostasis [68]. This occurred through the activation of sir-2.1, which led to an improvement in the disturbed balance between OXPHOS subunits encoded by mitochondrial DNA and nuclear DNA. This is related to the activation of UPRmt, which promotes longevity, and the subsequent translocation and activation of the FOXO transcription factor daf-16, triggering an antioxidant protective mechanism.
  • This point has also been added to the discussion (lines )
  1. Authors should provide future directions and describe key questions that can be addressed using C. elegans as a model to understand the role of phytochemical research in aging.
  • We improved the conclusion as following: In summary, the C. elegans nematode model offers a great advantage for mitochondrial research, especially by elucidating aging phenomena in a nutritional and environmental context. Because of the ease of handling this model organism, the high rate of mitochondrial maintenance, and the close monitoring of food sources, this ani-mal is an ideal candidate for monitoring much more than the aging process itself. Future nutritional scientists working with phenolic compounds and extracts from fruits and vegetables should consider C. elegans as a suitable model for their research pur-poses. Understanding interspecies transferability will play a major role in future re-search to find alternative models to classical animal models for drug or natural sub-stance research. Unveiling new effects of secondary plant compounds on metabolic pathways will be a key task for researchers. In addition, the potential application of these compounds as therapeutics will be of great interest. If the aging process cannot be halted, youthful physiology may be maintained into old age. Pharmaceutical treat-ment of age-related diseases could be supported with natural substances and the time of illness could be shortened.
  • This revised summary has also been added to the text (lines 387 to 400)

Reviewer 2 Report

This article presents information on the use of C. elegans in antioxidant, aging, and mitochondria research. The authors provide usefully and summarized information on the advantages of using C. elegans as a model, the techniques that could be performed on it, and the limitations of mitochondrial studies in this organism. However, the content is general and essential for the journal Biomolecules. Some topics need to be described in more detail, and it is suggested that the following comments be considered.

1.    There are some wording problems throughout the manuscript. It must be carefully reviewed.

2.    The topics of sections 4 and 5 only have a general idea of the advantages of the nematode in aging studies with phytochemicals and the mitochondrial approach; however, they do not give examples. It is unclear if it is a proposal or if there are already works that have done it. They would have to clear it up. Moreover, if articles already exist where anti-aging phytochemicals have been used with the mitochondria, a table describing them should be included. Also, at the end of section 5, research on the effect of phenolic compounds on mtDNA is cited; however, this work is not well described and is relevant to the section. Information on the specific effect of these compounds on mtDNA should be included.

3.    Section 5 contains information unrelated to the section or not well connected to the topic. Specifically, information on phytochemical delivery and biotransformation, as well as the definition of hormesis, is not connected to the rest of the section.

4.    Since the article refers to C. elegans, orthologous gene names should be included when mentioning the Nrf2 pathway (Section 5).

Author Response

Reviewer 2

This article presents information on the use of C. elegans in antioxidant, aging, and mitochondria research. The authors provide usefully and summarized information on the advantages of using C. elegans as a model, the techniques that could be performed on it, and the limitations of mitochondrial studies in this organism. However, the content is general and essential for the journal Biomolecules. Some topics need to be described in more detail, and it is suggested that the following comments be considered.

  1. There are some wording problems throughout the manuscript. It must be carefully reviewed.
  • Thank you for your detailed revision. We reviewed the paper and improved the wording problems.
  1. The topics of sections 4 and 5 only have a general idea of the advantages of the nematode in aging studies with phytochemicals and the mitochondrial approach; however, they do not give examples. It is unclear if it is a proposal or if there are already works that have done it. They would have to clear it up. Moreover, if articles already exist where anti-aging phytochemicals have been used with the mitochondria, a table describing them should be included. Also, at the end of section 5, research on the effect of phenolic compounds on mtDNA is cited; however, this work is not well described and is relevant to the section. Information on the specific effect of these compounds on mtDNA should be included.
  • We improved section four completely by adding a table with all classes of polyphenols, their prominent agents, and studied effects, as well as important literature. (Line 282)

Classes of polyphenols

Commonly used representatives

Studied effects

Literature

Flavone

Luteolin

·       sensitizes cancer cells to therapeutically induced cytotoxicity by suppressing cell survival pathways (PI3K/Akt, NF-κB, XIAB)

·       stimulates apoptosis pathways inducing the tumor suppressor p53

[99–103]

Baicalein

·       Anti-malignant potential through influence of several signaling cascades (MAPK, mTOR, PKB/Akt, PARP, MMP-2, MMP-9, caspase)

[104–108]

Apigenin

·       Induces intrinsic apoptosis pathways

·       Leads to downregulation of matrix metallopeptidases

·       Leads to downregulation of PI3K/Akt/ NF-κB signaling

[109–112]

Flavonol

Kaempferol

·       Inducement of apoptosis, cell cycle arrest at the G2/M phase

·       Downregulation of the epithelial-mesenchymal transition (EMT)-related markers, and PI3K/PKB signaling pathways

[113–116]

Myricetin

·       Therapeutic effects on atherosclerosis, thrombosis, diabetes, Alzheimer’s disease

·       Regulates MAPK, PI3K,/Akt/mTOR, IκB/NF –κB, AChE

·       Enhances immunomodulatory functions

[117–121]

Quercetin

·       Inhibits NF –κB

·       Activates SIRT1 via improving the NAD+ level

·       Inhibit α-glucosidase and increase adiponectin

·       Decreases the activity of inflammatory enzymes such as 5-LOX, 12-LOX, COX, NOS, and MPO

[122–125]

Flavanone

Hesperitin

·       Improve mitochondrial function through increasing complex function

·       Upregulate antioxidant levels (SOD, GPx, GR)by triggering PI3K/Akt pathway

·       Neuroprotection by regulating the TLR4/NF-κB signaling pathway

·       Augment the antioxidant cellular defenses via the ERK/Nrf2 signaling pathway

[126–129]

Naringenin

·       Inhibits TNF-α-induced TLR2 expression by inhibiting the activation of NF-κB and c-Jun NH2-terminal kinase pathways

·       Modulate the MAPK signaling pathway

·       Protective effect against LPS-induced injury

[130–132]

Flavanonol

Taxifolin

·       Anti-inflammatory through suppressing NF-κB activation

·       Hepatoprotective through reduced CD4+ and CD8+ T cells in injured liver tissue

·       Down-regulating the levels of TNF-α, COX-2

[133–135]

Engeletin

·       Mitigates Aβ1-42-induced oxidative stress and neuroinflammation through the Keap1/Nrf2 pathway.

·       Hepatoprotective through activation of PPAR-γ

·       Reduce NF-κB-dependent signaling

[136–138]

Anthocyanidin

Malvidin

·       Inhibit of IL-6, TNF-α, and IL-1β

·       Increase antioxidative enzymes (SOD, GPx)

·       Stimulate AMPK-mediated autophagy

[139,140]

Delphinidin

·       Increase expression of antioxidant protein HO-1 (Nrf2-related phase II enzyme heme oxygenase-1)

·       Modulate JAK/STAT3 and MAPKinase signaling to induce apoptosis

[141,142]

Flavan-3-ol

Epigallocatchin
gallate

·       Antiviral and antibacterial effects

·       Inhibit tumor necrosis factor-α (TNF-α)-induced production of monocyte chemoattractant protein-1 (MCP-1)

[143,144]

Isoflavone

Daidzein

·       Block the transcriptional activation of pro-inflammatory genes and decreased the mRNA level of Cxcl2 in TNFα-treated cells

·       Increase the AMPK phosphorylation followed by GLUT 4 translocation and glucose uptake

[145,146]

Genistein

·       Induce apoptosis through activation of caspase-1

·       Anti-proliferative effects through down-regulation of DNA methylation

·       Suppress Akt activity, promoting deactivation of NF-κB

[147–149]

  • In section 5 we included a figure of polyphenols which act on aging processes in cells (Line 375)
  •  
  • We improved the information on mtDNA as following: In addition, an influence on mtDNA by phenolic compounds and metabolites has also been observed in the past [158]. Resveratrol is a polyphenol found in grapes and red wine that possesses a number of biological activities, including anti-inflammatory and antioxidant activities [189]. Recent studies reported an increase of mtDNA quantity after the treatment with resveratrol [190,191]. Thus, may lead to an elevated level of mitochondrial biogenesis [192]. Other effects of polyphenols on mtDNA have also been demonstrated in disease models. As with resveratrol, incubation of human breast cancer cells MDA-MB-231 with this compound results in decreased levels of mtDNA [193]. This may be due to increased autophagy induced in response to mtDNA dam-age. Similarly, treatment with curcumin in human hepatoma cells HepG2 leads to in-creased damage to mtDNA [194]. This damage has been shown to trigger apoptosis in these cancer cell lines.
  • The important point has been discussed now (lines 363 to 374)
  1. Section 5 contains information unrelated to the section or not well connected to the topic. Specifically, information on phytochemical delivery and biotransformation, as well as the definition of hormesis, is not connected to the rest of the section.
  • Thank you for this review. We tried to improve this part and connected the relevance of the biotransformation to this topic as following: Before polyphenols can act as bioactive agents, they must reach the cells or compartments intended for them. Polyphenols undergo various biotransformations, not only by digestive enzymes but especially by the microbiota, changing the chemical structure and properties of the molecules consumed [164]. The human intestinal mi-crobiota consists of approximately 1012-1014 bacterial cells and is an extremely diverse entity involved in the digestion and fermentation of food components such as poly-phenols. In this context, the microbiota interacts closely with the immune system, making a balanced microbiota essential for maintaining a healthy state [165,166]. Ulti-mately, traceability in plasma reveals the availability of various phenolic acids after microbial remodeling. Therefore, it is important to note that the phenolic compounds circulating in the human system may differ greatly from those administered. Only those that reach the desired tissue can exhibit their bioactive functions.
    1. (Lines 314 to 325)
  • Also the hormesis section has been improved: In addition to direct antioxidant mechanisms, polyphenols have also been shown to initiate indirect mechanisms that promote innate detoxification pathways. One of these mechanisms is hormesis [182]. Hormesis refers to a biphasic, dose-dependent effect of bioactive substances. While high concentrations are considered toxic, moder-ate to low doses of exposure can be beneficial to health and activate cellular adaptive mechanisms [183]. A very prominent hermetic process in aging research is caloric re-striction. Reduced caloric intake has been shown to increase life expectancy in subjects [184]. In addition, it has been reported that phytochemicals can be neuroprotective by exhibiting hermetic processes through the involvement of various genes [175]. Heat shock proteins should be mentioned because temperature is an important hermetic factor. The induction of heat shock leads to the upregulation of heat shock proteins and chaperones. These types of proteins preserve the three-dimensional structure of pro-teins and assist newly synthesized proteins to fold correctly [185,186]. In C. elegans, the variation in hormesis effects was shown to be genetically determined. These results confirm that hormesis is formed by mechanisms that have been optimized during evolution [187]. A recent study in C. elegans showed that the phenolic acids protocate-chuic acid, gallic acid, and vanillic acid trigger the hormesis process [159].
  • This point has now been discussed (lines 342 to 358)
  1. Since the article refers to C. elegans, orthologous gene names should be included when mentioning the Nrf2 pathway (Section 5).
  • We improved this part as following: Rather, it is assumed that they activate among others the Keap1/Nrf2/ARE signaling pathway, which triggers a hormonal activation of phase II enzymes and thus strengthens the body's oxidative defense system [173,174]. In mammals, Drosophila melanogaster and C. elegans, detoxification pathways are tightly regulated. Thus, their basic activity is low. Contact with toxic xenobiotics or other oxidants then simultane-ously activates the expression of several genes through inducible transcription factors [175]. SKN-1 is a transcription factor orthologous to the mammalian Nrf2. Activated by various xenobiotics, oxidants, and electrophiles SKN-1 confers resistance by activating detoxification genes [176–179].
  • This point has also been added to the discussion (lines 333 to 341)

Round 2

Reviewer 1 Report

The authors have made all the suggested modifications to the original manuscript. Due to the incorporation of these changes, the revised manuscript has improved significantly. I am confident that this review article will help those studying the effect of phytochemicals on C. elegans biology. I support the publication of this manuscript.

Author Response

We thank Reviewer 1 for reviewing the manuscript and providing much assistance.

Reviewer 2 Report

This article presents information on the use of C. elegans in antioxidant, aging, and mitochondrial research. As the title suggests, it should include information on effects of polyphenols on aging and mitochondrial function in C. elegans, however, this information is not included. Furthermore, the topics are not described in detail nor discussed properly. Nervetheless, the authors provide useful information on the advantages of using C. elegans as a model of aging, the techniques that could be performed on it, and the advantages and limitations of mitochondrial studies in this organism.

1.    Redaction is still deficient, ideas need to be clearly connected. For instance, in section 2, advantages of working with C. elegans on aging research are mentioned, however, the idea of gene conservation on the nematode is not clearly linked to the previous and subsequent ideas, further, it is not stated that this fact is also an advantage of using C. elegans as a model.

2.    In section 2, age-1 mutant is not properly described and, thus, it is not possible to understand why it is evidence of the free radical hypothesis of aging. In general, ideas of the manuscript should be better discussed.

3.    Information on sterilization protocols on the nematode is not well integrated in section 2. It may be better linked to the rest of the section.

4.    There are several and relevant problems on section 3.

·      First, mitochondrial biogenesis is said to contribute to aging but not explained how.

·      DCT-1 role on aging, as well as the factors that lead to its activation, are not discussed, just mentioned.

·      Regarding bioenergetics section, it has concerning problems. This part appears to indicate that mitochondrial dysfunction is related with prolonged lifespan, while the paper the authors cite explains that mild-stress induced on larval stage leads to prolonged lifespan that is explained by enhanced basal and ATP-linked respiration.

·      Mitochondrial translation is said to be involved in aging but the mechanisms involved and references on this are missing.

·      Mitochondrial fusion and fission are said to be involved in aging but the role of these processes is not detailed.

·      AMPK and TORC1 information is not properly linked to the rest of the information on this section.

·      Further, section 3 title must be changed. An appropriate title could be “Mitochondria, aging and C. elegans”.

5.    A table summarizing studies of anti-aging properties of polyphenols, concerning mitochondrial function, was suggested to be added. Nevertheless, instead of that, the authors included a table on studies of several health-promoting effects of polyphenols, not related to the the topic of the manuscript.

6.    The paragraph added in section 4 (polyphenols in aging research), on C. elegans usage in pharmacological research, fits better in section 2, which talks about using C. elegans as a biological model.

7.    Section 5, which was expected to be a pivotal section of the paper as it concerns the effects of polyphenols on mitochondria, includes a few information on this and a lot of unrelated information (microbiota and hormesis). Furthermore, the few information on effects of polyphenols on mitochondria was not obtained in C. elegans, which was supposed to be the focus of the paper.

8.    In general, the paper is hard to be read because ideas are disconnected among them.

Author Response

Reviewer 2

This article presents information on the use of C. elegans in antioxidant, aging, and mitochondrial research. As the title suggests, it should include information on effects of polyphenols on aging and mitochondrial function in C. elegans, however, this information is not included. Furthermore, the topics are not described in detail nor discussed properly. Nervetheless, the authors provide useful information on the advantages of using C. elegans as a model of aging, the techniques that could be performed on it, and the advantages and limitations of mitochondrial studies in this organism.

  1. Redaction is still deficient, ideas need to be clearly connected. For instance, in section 2, advantages of working with C. elegans on aging research are mentioned, however, the idea of gene conservation on the nematode is not clearly linked to the previous and subsequent ideas, further, it is not stated that this fact is also an advantage of using C. elegans as a model.
  • Thank you for that important point. We transferred the informations about the overall homology bring it more to the overall advantages as of the nematode. It is now considered to the first part of section 2 and mentioned it as following: With the sequencing of the complete genome in 1998, C. elegans became accessible for molecular analyses. Of the ∼20,000 genes in C. elegans, an estimated that 15-30% are essential, but many have not yet been identified or characterized [24] It became clear that the similarity between the nematode and humans in terms of genetic background is remarkable. About 40% of the genes associated with human diseases are phyloge-netically identical to those in C. elegans, so-called homologs [25]. This fact makes the nematode a suitable model for understanding disease development.
  • The important point has been discussed now (lines 89 to 96)
  1. In section 2, age-1 mutant is not properly described and, thus, it is not possible to understand why it is evidence of the free radical hypothesis of aging. In general, ideas of the manuscript should be better discussed
  • We improved the information and hope that it is more understandable for readers.
  • Another mutant strain that exhibits significant lifespan extension, is the age-1 mutant [38,39]. Like daf-2, which also depends on DAF-16, age-1 is also integrated into the PI3K/Akt pathway and encodes a homolog of mammalian PI3K [40]. One of these nematode strains is called TJ401. This age-1 mutant strain showed an increased mean lifespan by 65 % compared to the wild-type nematode. Mutations, as in daf-2 and age-1 strains resulted in arrested larvae and forced larvae into the dauer stage. This increases nematode longevity and improves stress resistance [41]. The hyperresistance of these strains supports the free radical theory, one of the most important theories proposed by Denham Harman in 1956, since reactive oxygen species can modulate the import of DAF-16 into the nucleus, where it exerts its activity as a transcription factor [42,43]
  • The important point has been discussed now (lines 136 to 146)
  1. Information on sterilization protocols on the nematode is not well integrated in section 2. It may be better linked to the rest of the section.
  • We agreed with this point displaced the whole paragraph in an earlier part of this section
  • This whole paragraph can now been find in lines 113 to 129
  1. There are several and relevant problems on section 3.
  • First, mitochondrial biogenesis is said to contribute to aging but not explained how.
  • The relationship between mitochondria and the aging process has been described in several studies over the years. It is now believed that functional and dynamic changes in mitochondria trigger mitochondrial dysfunction and thus contribute to aging. Mi-tochondrial biogenesis is one of these changes. This mechanism is necessary for the enlargement of mitochondria by increasing their mass and number and is controlled by peroxisome proliferator-activated receptor-γ coactivator 1α (PGC-1α) [53]. It has been discussed that mitochondrial biogenesis may decrease due to an age-dependent de-crease in PGC-1α levels [54]. Although nematodes lack this protein, skn-1 also drives mitogenesis and thus has comparable functions to PGC-1α [55]. Regulation of mitochondrial biogenesis occurs through mitophagy, a selective form of autophagy.
  • This point has also been added to the discussion (lines 161 to 171)
  • DCT-1 role on aging, as well as the factors that lead to its activation, are not discussed, just mentioned.
  • In C. elegans, this process regulates mitochondrial content and nematode longevity. One of the key mediators of mitophagy and ensuring longevity under stress condi-tions, which is transduced to SKN-1 signaling, is DCT-1. dct-1 expression localized to the outer mitochondrial membrane is controlled in part by the FOXO transcription factor DAF-16 and is increased in the presence of low insulin/IGF-1 signaling [57]. It has been extensively studied that mitophagy mediated by dct-1 is involved in aging of C. elegans. With advancing age mitochondria accumulate in wild-type nematodes a de-ficiency of dct-1 and the autophagy key gene bec-1 recapitulates the effects of aging on mitochondrial mass in young adult animals. Induction of mitophagy was observed in long-lived daf-2 mutants. Impairment of mitophagy by knocking down dct-1, pink-1, and pdr-1 (the nematode Parkin homolog) significantly shortens the lifespan of daf-2 mutants. Indeed, dct-1 is transcriptionally induced under the control of skn-1 and daf-16 to remove dysfunctional mitochondria by mitophagy and coordinate mitochondrial biogenesis and mitophagy [57]. Mitophagy and mitochondrial biogenesis may work together to counteract the aging process [58].
  • This point has also been added to the discussion (lines 171 to 185)
  • Regarding bioenergetics section, it has concerning problems. This part appears to indicate that mitochondrial dysfunction is related with prolonged lifespan, while the paper the authors cite explains that mild-stress induced on larval stage leads to prolonged lifespan that is explained by enhanced basal and ATP-linked respiration.
  • We thank the reviewer to for this detailed observation of the bionergetic section. We changed the citation for a better understanding of this major point as following: With respect to bioenergetics as another conductor of aging, changes in basal and ATP-linked Oxygen Consumption Rate (OCR) at the critical third larval stage were found to be a potential predictor of lifespan extension in response to mitochondrial stress by RNAi. These changes most likely precede processes of reprogramming in fa-vor of longevity. It is likely that alterations in basal and ATP-linked OCR promote metabolic, genetic, and epigenetic reprogramming later in life, which are indeed caus-ally involved in longevity [59,60]. This leads to the finding that mild mitochondrial stress results in less profound changes in mitochondrial functional parameters [61]
  • This point has also been added to the discussion (lines 185 to 192)
  • Mitochondrial translation is said to be involved in aging but the mechanisms involved and references on this are missing.
  • We enlarged this part with more information and backround.
  • Another mechanism associated with the aging process is mitochondrial translation. This process consists of four phases: Ribosome initiation, elongation, termination, and recycling [62] The mitochondrial translation is carried out by mammalian mitochon-drial ribosomes (mitoribosomes). Their major task is to synthesize proteins essential for ATP production via oxidative phosphorylation. The mitochondrial translation is more similar to prokaryotic and differ from those of cytosplasmatic ribosomes [63]. This is due to the fact that mitochondrial protein synthesis requires a number of mito-chondrial factors at each stage [64]. Therefore, the main regulatory factor for initiation of the translation process MTIF2 has to be shown to play a role in pathological myo-cardial hypertrophy during aging and obesity [65]. Regarding the lack of MTIF2 in Saccharomyces cerevisiae results in Impaired mitochondrial protein synthesis affecting respiration [66]. Recent studies have shown that disruption of mitochondrial network homeostasis by blocking fusion or fission, combined with reduced mitochondrial translation, prolongs lifespan and stimulates the stress response [67]. The underlying reason for the increased lifespan is the influence of the primary lysosome biogenesis and autophagy transcription factor HLH-30/transcription factor EB (TFEB) [68,69]. This mainly suggests that mitochondrial dynamics are downstream of mitochondrial trans-lational stress to affect longevity, and that mitochondrial dysfunction transmits stress signals to lysosomes. These stimulate lysosome biogenesis and consequently longevity is promoted [67].
  • This point has also been added to the discussion (lines 192 to 210)
  • Mitochondrial fusion and fission are said to be involved in aging but the role of these processes is not detailed.
  • The mitochondrial network is maintained by mitochondrial fission and fusion. These processes coordinate a flexible and adaptable mitochondrial structure to the changing cellular environment. Studies in recent years have shown that mild mitochondrial dysfunction can delay aging and age-related loss of function in various animal models including mice, Drosophila melanogaster, and Caenorhabditis elegans [70–72]. In addition to the mitochondrial function of mitochondria, the shape of mitochondria is also affected by fission and fusion under these conditions. Espacially, for C. elegans there have been several orthologs for mitochondrial dynamics. There have been two orthologs for fusion proteins (FZO-1 and EAT-3) and three Fission Proteins (DRP-1, FIS-1 and FIS-2) [73] While there is an impact on aging, the role of mitochondrial dynamics in regulating lifespan is not well understood. Regarding studies of D. melanogaster and C. elegans mitochondrial fusion is associated with increased longevity and age correlates with fragmentation of the mitochondrial network [74–76].
  • This point has also been added to the discussion (lines 211 to 223)
  • AMPK and TORC1 information is not properly linked to the rest of the information on this section.
  • We linked that part to the mitochondrial dynamics section as followed:
  • Mitochondrial dynamics have been shown to be required for lifespan extension under various conditions of longevity, including target of rapamycin kinase complex 1 (TORC1)-mediated longevity, AMP-activated protein kinase (AMPK)-mediated lon-gevity, and in the presence of nutritional restriction [77,78]
  • This point has also been added to the discussion (lines 223 to 227)
  • Further, section 3 title must be changed. An appropriate title could be “Mitochondria, aging and C. elegans”.
  • We thank you for that suggestion. We changed the title of that section!
  • This point has also been added to the discussion (line 149)
  1. A table summarizing studies of anti-aging properties of polyphenols, concerning mitochondrial function, was suggested to be added. Nevertheless, instead of that, the authors included a table on studies of several health-promoting effects of polyphenols, not related to the topic of the manuscript.
  • We have included the table in Section 4 because this section talks about phytochemicals/polyphenols in general. The large number of polyphenols selected here have been excessively studied in previous studies and the positive influences listed have been generated as results. Among these are various effects on diverse diseases, immunological factors, mitochondrial parameters and aging-associated signaling pathways. Accordingly, in Section 5 we have chosen a simpler presentation of the effects of polyphenols on the organism and have created and included a pictorial graphic. In our opinion, this forms a simpler representation of the real effects, of polyphenols.
  1. The paragraph added in section 4 (polyphenols in aging research), on C. elegans usage in pharmacological research, fits better in section 2, which talks about using C. elegans as a biological model.
  • Thank you for this suggestion. We re-located this paragraph in section 2.
  1. Section 5, which was expected to be a pivotal section of the paper as it concerns the effects of polyphenols on mitochondria, includes a few information on this and a lot of unrelated information (microbiota and hormesis). Furthermore, the few information on effects of polyphenols on mitochondria was not obtained in C. elegans, which was supposed to be the focus of the paper.
  • In order to understand how polyphenols act in the organism, the reader must first understand the metabolism to which the substances are subjected. From this it must become clear to the reader that it may not be the primary substances whose effects have an influence on the aging process but their metabolites. Also the process of hormesis should be shown to the reader in relation to antioxidant phytochemicals. Indirect effects of hormesis on mitochondria are the focus of many studies nowadays
  • We improved the informations of polyphenols on mitochondria as following:
  • Another effect on aging by resveratrol in C. elegans has been demonstrated through di-rect interaction with mitochondrial respiration. By decreasing the activity of mito-chondrial respiration, resveratrol prolongs lifespan through a mechanism related to caloric restriction. In this case, sir-2.1 is increased after treatment of nematodes with resveratrol [218]. Influences on mitophagy were also observed. Catechinic acid resulted in a prolonged lifespan and a reduction in age-related behaviors by regulating genes associated with mitophagy pathways in C. elegans. By affecting bec-1 and pink-1, mito-chondrial phagocytosis is induced at early stages and lifespan may be affected [219]. The mechanism of mitophagy leads to the elimination of accumulated dysfunctional mitochondria, which can prolong lifespan. Epigallocatechin-3-gallate has been shown in recent studies to be able to restore mitochondrial function by increasing biogenesis in nematodes, thereby improving the redox status of nematodes [220]. Another inter-esting polyphenol that showed direct influences on the respiratory chain is oenothein B. This hydrolyzable polyphenol from the tannin group showed health-promoting ef-fects via isp-1, including reduction of ROS accumulation, improvement of motility flexibility, and aging pigments. The gene isp-1 encodes a subunit of mitochondrial complex III in C. elegans, coenzyme Q - cytochrome c reductase. [221].
  • This point has also been added to the discussion (lines 424 to 440)
  1. In general, the paper is hard to be read because ideas are disconnected among them.
  • We hope after the revision the topics are more connected among them and the paper is more understandable
